NAT10 inhibits ferroptosis and promotes the progression of renal clear cell carcinoma by regulating the NFE2L1-GPX4 signaling pathway

Tan Chao 1 2 3
Wang Yang 4
Zhang XinJie 2
Li ZiKang 2
Chen Shubo csb8160@126.com 1 2 3
1 Department of Urology, Hebei Medical University , Shijiazhuang , Hebei , China
2 Department of Urology, Xingtai People’s Hospital , Xingtai , Hebei , China
3 Department of Cell Laboratory, Hebei Provincial Key Laboratory of Portal Hypertension and Cirrhosis , Xingtai , Hebei , China
4 Department of Reproductive Medicine, Xingtai People’s Hospital , Xingtai , Hebei , China
Uversky Vladimir
Electronic publication date: 2025 Oct 31
Publication date: 2025
Volume: 13
Electronic Location ID: e20224
Received 2025 May 7; Accepted 2025 Sep 22
Copyright: ©2025 Tan et al.
Copyright year: 2025
Copyright holder: Tan et al.
License: This is an open access article distributed under the terms of the Creative Commons Attribution License, which permits unrestricted use, distribution, reproduction and adaptation in any medium and for any purpose provided that it is properly attributed. For attribution, the original author(s), title, publication source (PeerJ) and either DOI or URL of the article must be cited.
License URL: https://creativecommons.org/licenses/by/4.0/

Keywords: NAT10, Ferroptosis, ccRCC, NFE2L1, GPX4

Funding: Hebei Province 2024 Annual Medical Scientific Research Project Plan 20240450 This work was supported by Hebei Province 2024 Annual Medical Scientific Research Project Plan (Number: 20240450). The funders had no role in study design, data collection and analysis, decision to publish, or preparation of the manuscript.

==============================
Objective

Clear cell renal cell carcinoma (ccRCC) is a prevalent pathological subtype of renal cell carcinoma that arises from renal tubular epithelial cells. ccRCC has long been characterized by high mortality, and at present, surgical resection is the only curative treatment, but its effectiveness is low and survival rates are low. N-acetyltransferase 10 (NAT10) is a protein that acts as an acetyltransferase, has distinct catalytic and regulatory functions, and is involved in the progression of various cancers such as bladder cancer, hepatocellular carcinoma, and multiple myeloma. Ferroptosis is a novel form of programmed cell death that is iron-dependent and distinct from apoptosis, necrosis, and autophagy. Therefore, this study aimed to investigate the potential functions of NAT10 in this context, with a particular focus on its interactions and mechanisms involving ferroptosis.

Methods

In this study, immunohistochemistry, immunofluorescence, and Western blotting were utilized to evaluate the changes of NAT10 and ferroptosis-related marker proteins. Plasmid vectors were utilized to establish stable ccRCC cell lines with NAT10 overexpression and knockdown. Cell proliferation, scratch, and invasion assays were conducted to assess the impact of NAT10 modification on cell proliferation and migration capabilities. Meanwhile, remodelin hydrobromide (HBr) treatment was given to ccRCC cells to observe the variations in cell proliferation ability and ferroptosis marker proteins.

Results

Our study demonstrated that NAT10 can enhance the malignant biological behavior of ccRCC, while simultaneously inhibiting markers associated with ferroptosis.

Conclusions

Therefore, this study suggests that NAT10 may exert a pro-cancer effect by modulating the nuclear factor erythroid-2, like-1(NFE2L1)-glutathione peroxidase-4(GPX4)signaling pathway in ccRCC, indicating its potential as a new therapeutic target for this malignancy.

Introduction

Clear cell renal cell carcinoma (ccRCC) is a malignant tumor arising from the epithelial cells of the proximal convoluted tubules of the kidney, accounting for approximately 85% of all malignant renal tumors. Its incidence rate ranks among the top ten in both male and female malignant tumors (Bian, Fan & Xie, 2022; Tan et al., 2023; Charbonneau et al., 2022). ccRCC has an insidious onset, a high metastasis rate, and a poor prognosis. However, it is insensitive to radiotherapy and chemotherapy. Currently, the main treatment method is surgery; however, 20% to 40% of patients still suffer from local recurrence or distant metastasis after surgery. Consequently, to enhance the prognosis of patients with ccRCC, it is extremely important to search for biomarkers for the prediction and treatment of ccRCC.

N-acetyltransferase 10 (NAT10) is a nuclear protein that is widely expressed across various cell types. It comprises 1,025 amino acids and features an acetyltransferase domain along with a lysine-rich carboxy-terminal region (Shi et al., 2023). In normal tissues, NAT10 is predominantly localized in the nucleolus; however, in tumor cells, it exhibits aberrant expression patterns within the nucleus, cytoplasm, and cell membrane (Wang et al., 2022a). Recent investigations have indicated that NAT10 participates in several physiological and pathological processes in the human body, including the regulation of mRNA stability and translation efficiency, modulation of telomerase activity, control of cell cycle progression, and involvement in cellular proliferation, migration, DNA damage response mechanisms, and epithelial-mesenchymal transition (EMT) processes (Liao et al., 2023; Dalhat et al., 2022; Tan et al., 2018). Although studies have shown that upregulation of NAT10 promotes EMT, thereby promoting metastasis in hepatocellular carcinoma (Guo et al., 2024), its role in kidney cancer, especially ccRCC, is still poorly understood. Recent studies have shown that N-acetyltransferase 10 (NAT10) and its mediated RNA N4-acetylcytidine (ac4C) modification are significantly upregulated in clear cell renal cell carcinoma (ccRCC) and are associated with poor prognosis in patients. On the one hand, NAT10 (activated by HIF-1α transcription) enhances the stability of NFE2L3 mRNA through catalyzing ac4C modification, thereby upregulating LASP1 expression, activating the AKT/GSK3β/β-catenin signaling pathway, and promoting tumor cell proliferation, migration, in vivo growth, and metastasis in ccRCC (Sun et al., 2025). On the other hand, NAT10 also upregulates ANKZF1 protein expression by mediating the ac4C modification of ANKZF1 mRNA; ANKZF1 inhibits the cytoplasmic retention of YAP1 by competitively binding with YWHAE, promoting YAP1 nuclear translocation and its transcriptional activity, and ultimately activates pro-lymphangiogenic factors, driving lymphangiogenesis and tumor progression in ccRCC. These results reveal that NAT10/ac4C modifications play a core oncogenic role in ccRCC tumor growth and metastasis (including lymphatic metastasis) by regulating two key signaling axes: NFE2L3-LASP1-AKT/GSK3β and ANKZF1-YWHAE-YAP1, providing new potential avenues for targeted interventions (Miao et al., 2024).

Ferroptosis is a novel form of programmed cell death that is iron-dependent and distinct from apoptosis, necrosis, and autophagy (Dixon & Olzmann, 2024; Liang, Minikes & Jiang, 2022; Gao et al., 2022). This process can be activated under various acute and chronic pathophysiological conditions that are associated with cellular damage. The primary mechanism underlying ferroptosis involves lipid peroxidation of highly expressed unsaturated fatty acids within the cell membrane, catalyzed by divalent iron or lipoxygenase, ultimately resulting in cell death (Liu, Kang & Tang, 2022; Yang & Yang, 2022). Furthermore, ferroptosis is characterized by reduced expression levels of antioxidant enzymes, particularly glutathione (GSH) and glutathione peroxidase 4 (GPX4) (Wang et al., 2025). This phenomenon may also be triggered by cysteine depletion (the oxidized form of cysteine) (He et al., 2022). The consumption of cysteine or inhibition of GPX4 leads to the accumulation of iron-dependent lipid peroxides in the plasma membrane and other cellular compartments, culminating in alterations in membrane permeability that induce cell death (Yang & Yang, 2022). There are studies showing that nuclear factor erythroid-2, like-1 (NFE2L1) prevents ferroptosis by promoting the expression of GPX4 (Forcina et al., 2022). The principal characteristics of ferroptosis include: in terms of cell morphology, it can lead to smaller mitochondria, increased membrane density, reduced cristae, and no significant changes in nuclear morphology. Regarding cell components, ferroptosis is characterized by enhanced lipid peroxidation and elevated reactive oxygen species (ROS) (Lai et al., 2022). Previous studies have confirmed that ferroptosis is associated with various diseases such as cancer, neurodegeneration, and ischemia-reperfusion injury. In the context of cancer, ferroptosis is involved in regulating the occurrence, development, invasion, metastasis, and treatment resistance of different types of cancer. At the same time, studies have shown that there is a close connection exists between NAT10 and ferroptosis during tumor progression; for example, NAT10 plays a critical role in colon cancer development by affecting ferroptosis-suppressor-protein 1 (FSP1) mRNA stability and ferroptosis, suggesting that NAT10 could be a novel prognostic and therapeutic target in colon cancer (Zheng et al., 2022). m6A-driven NAT10 translation facilitates fatty acid metabolic rewiring to suppress ferroptosis and promote ovarian tumorigenesis by enhancing Acyl-CoA thioesterase 7 (ACOT7) mRNA acetylation (Liu et al., 2024). However, too date, there have been no published reports regarding the role and molecular mechanisms of NAT10 and ferroptosis in clear cell renal carcinoma. Therefore, this study aimed to investigate the potential functions of NAT10 in this context, with a particular focus on its interactions and mechanisms involving ferroptosis (Mishima & Conrad, 2022; Strauss et al., 2022; Green et al., 2022).

Materials and Methods

Tissue processing

In this study, 30 cases of primary renal clear cell carcinoma and distal normal tissue adjacent to the cancer (more than five cm away from the cancer tissue) and 30 cases of normal renal tubular epithelial tissue were collected as controls. The samples were provided by the Xingtai People’s Hospital. All tissues have been pathologically diagnosed as renal clear cell carcinoma, and their diagnosis is carried out according to the standards of the World Health Organization, and is evaluated by senior experts from the pathology center. All patients had primary lesions and did not undergo preoperative radiotherapy, chemotherapy, or other anticancer treatments. The collection of tumor specimens received informed consent from patients and their families. Prior to study commencement, all experimental procedures and research objectives were thoroughly explained to participating patients and their families. Each participant voluntarily enrolled in the trial with documented informed consent agreements. Written informed consent was obtained from participants. Strict confidentiality protocols safeguarded personal information, while participants retained the unequivocal right to withdraw from the research at any stage without consequences. The study was approved by the Xingtai People’s Hospital Ethics Committee (2023 [035]).

Immunohistochemistry

The paraffin sections were placed overnight in a 65 °C oven, followed by gradient dewaxing and hydration. Then, 3% H2O2 solution was added and incubated at room temperature for 15 min to block endogenous peroxidase activity. The slices were placed in citrate buffer and subjected to microwave heating to repair the antigens. Normal sheep serum (5%) was added for blocking and incubated at room temperature for 15 min. Subsequently, immunohistochemistry and immunofluorescence were used to detect the target genes.

Immunofluorescence

Tissue sections were fixed in pre-cooled methanol:acetone (1:1) for 15 min at 4 °C, followed by permeabilization with 1% Triton X-100 in PBS for 10 min at room temperature to enable antibody penetration. After washing and drying, incubate overnight with NAT10, PTGS2, CHAC1, NFE2L1, and GPX4 (1:1000) antibodies in at 4 °C refrigerator. After primary antibody incubation, tissue sections were post-fixed with 4% paraformaldehyde (PFA) for 20 min at room temperature to stabilize antibody-antigen complexes, followed by cryoprotection in 15% sucrose/PBS overnight at 4 °C for optimal structural preservation during confocal imaging. The glass slides were cleaned with deionized water and the slices were analyzed using laser confocal scanning.

For immunohistochemistry (IHC) and immunofluorescence (IF) analysis, only tubular epithelial cells (the cell type of origin of clear cell renal cell carcinoma) were stained and evaluated, excluding interstitium, inflammation, and glomerular cells. Five fields of view (40x magnification, 0.25 square millimeters per field of view) were randomly selected for each sample, and each field counted ≥ 200 cells (total ≥ 1,000 cells per sample). The staining intensity score is as follows: 0 (none), 1 (weak), 2 (moderate), or 3 (strong). The formula for calculating the percentage of positive cells is: percentage of positive = number of positive cells/amount of total number of cells × One hundred.

A composite score (range: 0–300) is obtained by multiplying the intensity score by the percentage of positivity. Adjacent normal kidney tissue with score matched to patient was standardized using the following ratio: Standardized score = tumor composite score/adjacent normal tissue composite score.

Two blinded pathologists performed the score. All immunofluorescence images contain scale bars (specified in the legend) and pseudocolor labels: NAT10 (red), GPX4 (green), DAPI (blue).

Quantitative real-time polymerase chain reaction

Thaw the frozen tumor tissue stored at −80 °C slowly on ice to avoid nucleic acid degradation caused by repeated freeze-thaw cycles. Place the tissue pieces into a pre-cooled mortar, add liquid nitrogen, and rapidly grind them into a fine powder. Transfer the powder to a centrifuge tube. Wash 110–120 supernatant to a new EP tube. Add 200 µL chloroform, vigorously vortex for 15 s, and incubate at room temperature for 5 min; centrifuge at 4 °C for 15 min. Transfer the upper aqueous phase to a new EP tube, add an equal volume of isopropanol, and mix thoroughly. Allow to stand at room temperature for 10 min; centrifuge at 4 °C for 10 min. Discard the supernatant and wash the RNA pellet twice with one mL of 75% ethanol; centrifuge at 4 °C for 5 min, discard the supernatant, and air-dry at room temperature for 5 min. Dissolve the RNA in 50 µL DEPC-treated water for reverse transcription, and store at −80 °C. Total RNA was isolated from the cultured cells using TRIzol reagent (Promega, Madison, WI, USA). After reverse transcription using a reverse transcriptase kit (TaKaRa, Shiga, Japan), we performed real-time PCR for NAT10,NFE2L1, and GPX4 using SYBR green (TaKaRa, Shiga, Japan) on a Bio-Rad PCR system (Bio-Rad, Hercules, CA, USA). The primer sequences used are listed below. Gene expression levels were calculated using the 2−ΔΔCT method, and the data were normalized to GAPDH. The conditions are set as follows: initial denaturation at 95 °C for 10 min; denaturation at 95 °C for 10 s; annealing at 60 °C for 20 s and extension at 72 °C for 35 s. Primer information is provided in Table 1, and additional data are summarized in Table 1. The experiment was repeated three times for each group (Table 1).

Table 1 Primer sequences.

Primer sequences used for quantitative real-time polymerase chain reaction (qRT-PCR) to detect the expression of NAT10, NFE2L1, and GPX4

Gene	RefSeq	Forward	Reverse	
NAT10	NM_024662.3	5′ TAATACGACTCACTATAGGG 3′	5′ TAGAAGGCACAGTCGAGG 3′	
NFE2L1	NM_001145933.1	5′ TAATACGACTCACTATAGGG 3′	5′ TAGAAGGCACAGTCGAGG 3′	
GPX4	NM_001367864.1	5′-GGAAGCTGCAAGATGTCGAC-3′	5′-CAGCCACACCACATCATTCC-3′	

Cell transfection

On the day before transfection, logarithmic ACHN and Caki1 cells were incubated overnight in a 6-well plate at a density of 3 × 105/well, and antibiotic-free growth medium (2.5 mL) was added to each well. After the cell growth density reached 60%, the growth medium and add 1.5 mL of serum-free growth medium. ACHN and Caki1 cells were transfected with sh-NAT10 (50 nM) and OE-NAT10 overexpression plasmids. The sh-NAT10 constructs were based on the pLKO.1 backbone (Addgene, Watertown, MA, USA) and co-transfected with psPAX2 and pMD2.G packaging plasmids, while the NAT10 overexpression plasmid was constructed in the pcDNA3.1 backbone. Cells cultured in a carbon dioxide incubator at 37 °C for 5 h. The culture medium containing serum was changed after 48 h, and cells were collected after transfection for 36 h. The shRNA sequences targeting NAT10 were as follows:

shRNA-NAT10 (OBiO, Y29989): GAGATGTATTCACGGAATATG;

shRNA-NAT10 (OBiO, Y29990): GCAATTGTACACAGTGACTAT;

shRNA-NAT10 (OBiO, Y29991): CGGCCATCTCTCGCATCTATT;

Negative control (GL427NC2): CCTAAGGTTAAGTCGCCCTCG).

Western blotting

Caki1 cells at the logarithmic growth phase were lysed using radioimmunoprecipitation assay (RIPA) buffer. The total protein was collected by centrifugation at 12,000 rpm for 5 min at 4 °C. The protein samples were separated by 10% sodium dodecyl sulfate polyacrylamide gel electrophoresis (SDS-PAGE) and transferred onto a polyvinylidene difluoride (PVDF) membrane. The membrane was blocked with 5% skim milk for 2 h at room temperature and then washed. Subsequently, the membrane was incubated with primary antibody (diluted at 1:500) overnight at 4 °C, followed by washing. Then, it was incubated with the corresponding horseradish peroxidase (HRP)-conjugated secondary antibody (diluted at 1:5000) for 1 h at room temperature. Protein bands were visualized using an ECL kit (P0018FS), and the grayscale values were analyzed with Image J software. GAPDH was used as the internal control to calculate the relative expression levels of NFE2L1, GPX4, ACSL4, and SLC7A11 proteins.

CCK-8 experiment

Cells were cultured at a density of 70% and collected for preparation as a 1 ×104/mL cell suspension. A 96 well plate with 100 cell suspensions were added per well µL. Add 10 µL at 0, 12, 24, and 48 h respectively CCK-8 solution, incubate for 2 h, and an enzyme-linked immunosorbent assay was used to detect the absorbance value at a wavelength of 450 nm.

Cell scratch

Logarithmic growth phase cells were inoculated onto a 6-well plate, and when the cells reached 80%, a 200 µm gun was used to scratch the cover of the 6-well plate vertically. The culture medium was removed, rinsed with PBS, and fresh culture medium was added. After 36 h of cultivation, photos were taken for analysis.

Quantification of iron, malondialdehyde and reactive oxygen species

Iron: measured using an iron assay kit (I291-DOJINDO). The tissue/cell lysate was mixed with a chromogen and incubated at 25 °C for 30 min with absorbance reading at 593 nm. The results are normalized to protein concentration (µmol/g).

Malondialdehyde: detected by thiobarbituric acid active substance (TBARS) assay (AKFA013M). The sample was reacted with TBA at 95 °C for 60 min, cooled, and absorbance (nmol/mg protein) was measured at 532 nm.

ROS: cells were stained with 10 µM DCFH-DA (S0033S) at 37 °C for 30 min. Fluorescence intensity was determined by flow cytometry. Data are presented as fold changes vs. controls.

Transwell uses transwell to detect cell invasion

The diluted Matrigel matrix (1:8 ratio) was added to the upper chamber of the Transwell insert and incubated at 37 °C for 30 min to allow polymerization. Subsequently, 600 µL of complete medium containing 20% fetal bovine serum (FBS) was added to the lower chamber of a 24-well plate. Cells were serum-starved in serum-free medium for 24 h at 37 °C, trypsinized, and resuspended in serum-free medium at a density of 5 × 104 cells/mL. Then, 100 µL of the cell suspension was carefully added to the pre-hydrated upper chamber. After 24 h of incubation, the non-invading cells in the upper chamber were gently removed with a cotton swab. The membrane was washed twice with phosphate-buffered saline (PBS), and the invaded cells on the lower surface were fixed with 95% ethanol and stained with 0.1% crystal violet dye (Solarbio, Beijing, China) for 20 min. The number of invaded cells was counted in five randomly selected fields under an inverted microscope at ×400 magnification.

Statistics

Data are expressed as mean ± standard error of the mean (SEM). The Student’s t-test was used for comparisons between the two groups. For multiple comparisons, data were analyzed using one-way ANOVA followed by Tukey’s post-hoc tests P < 0.05.

Results

Expression changes of NAT10 in renal clear cell carcinoma tissue

To preliminarily explore the role of NAT10 in renal clear cell carcinoma and analyze its expression in renal clear cell carcinoma tissue, NAT10 was found to be significantly overexpressed in renal clear cell carcinoma tumor tissue (Figs. 1A–1C). Further studies have shown that the expression of NAT10 in metastatic tumors was higher than that in non-metastatic tumors (Figs. 1D–1E). Meanwhile, we carried out PCR verification of the tissues positive for NAT10 immunohistochemical detection, and the results indicated that the expression level of NAT10 mRNA was significantly increased in NAT10 positive tissues (Fig. 1F). The analysis of the prognosis of patients revealed that the survival rate of patients with elevated expression of NAT10 was poor (Fig. 1G).

Figure 1 Expression changes of NAT10 in renal clear cell carcinoma tissue.

(A, B) Immunohistochemistry and (A, C) immunofluorescence were used to detect the expression of NAT10 in normal tissues and tumor tissues. NAT10 (red), GPX4 (green), and nuclei (DAPI, blue), Scale: 2 0 µM. (D) RT-PCR was used to detect the expression of NAT10 in normal tissues and tumor tissues. (E) RT-PCR was used to detect the expression of NAT10 in metastatic and non-metastatic tumor tissues. (F) RT-PCR was employed to detect the expression of NAT10 in tumor tissues which were identified as NAT10 positive and negative by immunohistochemistry. (G) The relationship between NAT10 and the prognosis survival rate of patients with clear cell renal cell carcinoma. Statistical significance: *** P < 0.001 by one-way ANOVA followed by Tukey’s post-hoc tests.

NAT10 promotes the malignant biology of renal clear cell carcinoma

NAT10 promotes the malignant biological behavior of renal clear cell carcinoma. We respectively established NAT10 overexpression and knockdown cell lines (Figs. 2A–2C). Subsequently, the proliferative capacity of transfected cells was evaluated using CCK-8. The results demonstrated that overexpression of NAT10 significantly enhanced the proliferative capacity of renal clear cell carcinoma cells, whereas knockdown led to the opposite outcome (Figs. 2D–2E). Simultaneously, cell scratch and cell invasion assays were conducted to evaluate the migratory and invasive abilities of the tumor cells. These findings demonstrated that NAT10 promoted cell migration (Figs. 2F, 2G, 2H) and invasion (Figs. 2F, 2I, 2J). These results indicate that NAT10 is closely associated with the occurrence and progression of renal clear cell carcinoma.

Figure 2 NAT10 promotes the malignant biology of renal clear cell carcinoma.

(A) RT-PCR was used to detect the expression of NAT10 in different renal clear cell carcinoma cell lines. (B) shNAT10 silenced the expression of NAT10 in ACHN cells. (C) OE-NAT10 overexpression of NAT10 in Caki1. (D) CCK-8 was used to evaluate the changes in cell proliferation ability of ACHN cells after silencing the expression of NAT10. (E) CCK-8 was used to evaluate the changes in cell proliferation ability after overexpression of NAT10 in Caki1. (F, G, H) Cell scratch and (F, I, J) cell invasion assays were used to evaluate the migration and invasion abilities of ACHN cells and Caki1 cells after the change of NAT10, respectively. Statistical significance: *P < 0.05, **P < 0.01, *** P < 0.001 by one-way ANOVA followed by Tukey’s post-hoc tests.

Bioinformatics identification of ferroptosis-related genes and pathways enrichment associated with NAT10 in ccRCC

ccRCC transcriptome analysis from GEO datasets (GSE53757, GSE66271) revealed 5,888 degs between NAT10 high and low expression tumors, of which 3,198 genes were up-regulated and 2,690 genes were down-regulated. Cross-over of FerrDb’s ferroptosis-related down-regulated genes identified 177 overlapping candidate genes, many of which (e.g., GPX4, SLC7A11, FTH1) are key regulators of ferroptosis. Hierarchical clustering showed that these ferroptosis-related transcripts were significantly inhibited in the NAT10 high expression group. KEGG enrichment analysis showed that DEGs (differentially expressed genes) were significantly enriched in glutathione metabolism, oxidative stress response, NFE2L1-mediated antioxidant pathway and ferroptosis signaling, indicating that the NFE2L1-GPX4 axis is a central regulatory mechanism. These bioinformatics findings suggest that high NAT10 expression in ccRCC is associated with ferroptosis inhibition through transcriptional regulation of key antioxidant and iron metabolism pathways (Figs. 3A–3C).

Figure 3 Bioinformatics identification of ferroptosis-related genes and pathways enrichment associated with NAT10 in ccRCC.

(A) Volcano plot showing DEG (|log2FC| > 0.7, adjusted p < 0.05) between ccRCC tumors with high and low NAT10 expression. Red dots: genes were significantly up-regulated; Blue dots: significantly down-regulated genes; Gray dot: not significant. (B) Heat map of 177 overlapping genes between NAT10-related DEGs and ferroptosis-related down-regulated genes in clear cell renal cell carcinoma. Red: high expression; Blue: low expression; Values were standardized across samples against z-scores. (C) Bubble map of KEGG pathway enrichment of DEGs in renal clear cell carcinoma. Point size represents the number of genes. Colors indicate adjusted p-values. Notable enriched pathways included glutathione metabolism, oxidative stress response, NFE2L1-mediated antioxidant pathways, and ferroptosis signaling.

NAT10 regulates the expression of ferroptosis-related proteins

To evaluate the impact of NAT10 on ferroptosis, we examined the molecular changes associated with ferroptosis after changes in NAT10. We investigated the expression of NFE2L1 and GPX4 in both normal and tumor tissues. The results demonstrated that the protein (Figs. 4A–4C) and mRNA (Figs. 4D, 4E) levels of NFE2L1 and GPX4 were significantly increased in tumor tissues, whereas the ferroptosis marker proteins PTGS2, and CHAC1 were expressed at low levels in tumor tissues (Figs. 4F–4H), however, SLC7A11 exhibited significant overexpression in tumor tissues (Fig. 4I). Malondialdehyde, a product of membrane lipid peroxidation, is positively associated with ferroptosis in cells exposed to iron. Our findings suggest that upon overexpression of NAT10, the expression of iron (Figs. 4J–4K) and MDA (Figs. 4L–4M) was significantly decreased, while with the knockdown of NAT10, the expression of iron and MDA increased significantly the same time, while the expression of ROS increased with the increase of NAT10 (Figs. 4M–4Q).

Figure 4 NAT10 regulates the expression of ferroptosis-related proteins.

(A, B, C) Immunohistochemical assessment of the expression of GPX4 and NFE2L1 in normal tissues and renal clear cell carcinoma tumor tissues. (D) RT-PCR was used to evaluate the expression of GPX4 in normal tissues and tumor tissues. (E) RT-PCR was used to evaluate the expression of NFE2L1 in normal tissues and tumor tissues. (F, G, H, I) Immunofluorescence was used to evaluate the expression of ferroptosis marker proteins in normal tissues and tumor tissues. (J) The expression of Iron in ACHN cells was decreased after silencing NAT10. (K) The expression of Iron after overexpression of NAT10 in Caki1. (L) The expression of MDA in ACHN cells was decreased after silencing NAT10. (M) The expression of MDA after overexpression of NAT10 in Caki1. (N, O) The expression of ROS in ACHN cells was decreased after silencing NAT10. (P, Q) The expression of ROS after overexpression of NAT10 in Caki1. Statistical significance: ∗ P < 0.05, ∗∗ P < 0.01, ∗∗∗ P < 0.001 by one-way ANOVA followed by Tukey’s post-hoc tests.

NAT10 suppresses ferroptosis via the GPX4-NFE2L1 signaling pathway

We determined the expression of GPX4 and NFE2L1 in shNAT10-ACHN and OENAT10-Caki1 cells. The results indicated that both at the mRNA (Figs. 5A, 5B) and protein (Figs. 5C–5F) levels, GPX4 and NFE2L1 decreased upon the silencing of NAT10 and increased with the overexpression of NAT10.ACHN and Caki1 cells treated with DMSO and the NAT10-specific inhibitor remodelin hydrobromide (HBr), respectively. Finally, after overexpressing NFE2L1 in ACHN and Caki1 cells and then treating the cells with HBr, the results suggested that the inhibition of GPX4 expression by HBr was partially reversed (Figs. 6A, 6B, 6E, 6F, 6G). CCK-8 results demonstrated that HBr inhibited the proliferation of ACHN and Caki1 cells. Nevertheless, overexpression of NFE2L1 partially neutralized the inhibitory effect of HBr on cell proliferation (Figs. 6C, 6D). MDA and Iron displayed the same trend (Figs. 6H, 6I). Finally, the expression of the ferroptosis marker proteins was examined. These results revealed that HBr increased the expression of PTGS2, and CHAC1. However, when NFE2L1 was overexpressed, the expression of, PTGS2, and CHAC1 weakened in ACHN (Fig. 6J) and Caki1 (Fig. 6K). The trend of SLC7A11, however, exhibited an inverse pattern compared to the aforementioned markers. These results suggest that NAT10 inhibits ferroptosis through the GPX4-NFE2L1 signaling pathway, thereby promoting the malignant progression ofccRCC.

Figure 5 NAT10 regulates the expression of GPX4 and NFE2L1.

(A) RT-PCR was used to evaluate the expression of GPX4 and NFE2L1 in ACHN cells after silencing NAT10. (B) RT-PCR was used to evaluate the expression of GPX4 and NFE2L1 in Caki1 cells after overexpression NAT10. (C, D) Western Blotting evaluated the expression of GPX4 and NFE2L1 in Caki1 cells. (E, F) IF evaluated the expression of GPX4 and NFE2L1 after NAT10 silencing and overexpression. Statistical significance: ∗ P < 0.05, ∗∗ P < 0.01, ∗∗∗ P < 0.001 by one-way ANOVA followed by Tukey’s post-hoc tests.

Figure 6 NAT10 suppresses ferroptosis via the GPX4-NFE2L1 signaling pathway.

(A) RT-PCR was used to evaluate the expression of GPX4 in different treatment groups of ACHN cells. (B) RT-PCR was used to evaluate the expression of GPX4 in different treatment groups of Caki1. (C) The proliferation ability of ACHN cells in different treatment groups was evaluated by CCK-8. (D) The proliferation ability of Caki1 cells in different treatment groups was evaluated by CCK-8. (E, F, G) IF evaluate the expression of GPX4 in cells of different treatment groups. (H) The expression of MDA in ACHN and Caki1 cells of different treatment groups. (I) The expression of iron in ACHN and Caki1 cells of different treatment groups. (J, K) Changes of ferroptosis marker proteins in ACHN cells of different treatment groups. (L, M) Changes of ferroptosis marker proteins in Caki1 cells of different treatment groups. Statistical significance: ∗ P < 0.05, ∗∗ P < 0.01, ∗∗∗ P < 0.001 by one-way ANOVA followed by Tukey’s post-hoc tests.

GPX4 knockdown reverses the ferroptosis resistance induced by NAT10 overexpression

To further confirm whether GPX4 is a necessary downstream factor of NAT10, we conducted rescue experiments in Caki1 cells with NAT10 overexpression by using RSL3 to inhibit the expression of GPX4. Western blot and qRT-PCR analyses showed that after overexpressing NAT10, the protein and mRNA levels of GPX4 and SLC7A11 increased, while the protein and mRNA levels of ACSL4 decreased, indicating that NAT10 overexpression can inhibit ferroptosis. However, when a GPX4 inhibitor was added to NAT10-overexpressing Caki1 cells, the protein and mRNA levels of GPX4 and SLC7A11 decreased, while the protein and mRNA levels of ACSL4 increased, demonstrating that the ability of NAT10 to inhibit ferroptosis was significantly weakened when the expression of GPX4 was suppressed (Figs. 7A–7B). CCK-8 assays showed that after the inhibition of GPX4 expression, the enhanced proliferation caused by NAT10 overexpression was significantly reduced (Fig. 7C), and the cellular MDA levels increased (Fig. 7D). These results prove that NAT10 inhibits ferroptosis in ccRCC cells through GPX4.

Figure 7 GPX4 knockdown reverses the ferroptosis resistance induced by NAT10 overexpression.

(A) qRT-PCR and (B, C) Western blot confirmed that the inhibition of GPX4 expression weakened the role of NAT10 in suppressing ferroptosis. (D) CCK-8 assays indicated that inhibiting GPX4 expression reduced the proliferation advantage of NAT10 overexpression. (E) Detection of MDA expression proves that inhibiting GPX4 expression promotes ferroptosis in cells overexpressing NAT10. Statistical significance: ∗ P < 0.05, ∗∗ P < 0.01, ∗∗∗ P < 0.001 by one-way ANOVA followed by Tukey’s post-hoc tests.

Discussion

The incidence of renal cell carcinoma (RCC) has increased continuously, reaching as high as 3%–5% in all adult malignant tumors, of which clear cell RCC is the most common and aggressive pathological subtype (Choueiri et al., 2021; Motzer et al., 2021; Choueiri et al., 2023). Therefore, there is an urgent need to further describe the mechanism of metastasis and progression of renal clear cell carcinoma in order to obtain more effective anti-tumor therapies. NAT10 exhibits significant translocation changes in tumor cells and is positively correlated with EMT. NAT10 promotes tumor cell invasion and metastasis by inducing EMT (Yang et al., 2021; Zheng et al., 2022; Jin et al., 2022), but the molecules and pathways involved are not well understood due to limited research and require further exploration. Therefore, elucidating the molecular mechanism by which NAT10 invasion and metastasis in ccRCC is of significance importance for future clinical management. Our findings reveal that NAT10 is strongly expressed in renal clear cell tumor tissues and is closely related to tumor metastasis and the unfavorable prognosis of patients with renal clear cell carcinoma. This suggests that NAT10 likely plays a vital role in promoting malignancy in renal clear cells. In addition, in this study, we analyzed 30 paired ccRCC tumor and adjacent non-tumor tissue samples. The patients included early-stage (stage I–II, 65%) and late-stage (stage III–IV, 35%) disease, with approximately 20% exhibiting high-grade tumors. Although these cases allowed us to preliminarily identify associations between NAT10 expression, ferroptosis markers, and patient prognosis, the limited sample size and uneven clinical pathology distribution inevitably restricted the robustness of stratified analyses. For instance, most samples lacked critical molecular characteristics, such as VHL mutation status, which significantly influences iron metabolism in ccRCC. These limitations may affect our interpretation of the results. Therefore, future studies on a larger and more comprehensively annotated patient cohort are crucial to validate our findings and determine whether NAT10 expression correlates with specific clinical pathological features (e.g., tumor grade, stage, or molecular subtype). Regarding the role of NAT10 in promoting cancer, we conducted further studies to determine its role in promoting cancer. It was clearly found that in renal clear cell carcinoma ACHN and Caki1 cells, when the expression of NAT10 was silenced, the proliferation, migration and invasion capabilities of the tumor were significantly reduced. Conversely, after overexpression of NAT10, the above-mentioned malignant behavioral abilities were significantly enhanced. This strongly indicates that NAT10 plays a crucial role in promoting cancer in renal clear-cell carcinoma and deserves further exploration.

In the exploration of malignant tumors, ferroptosis is involved in regulating the occurrence, development, invasion, metastasis, and treatment resistance of various types of cancers (Mou et al., 2019; Zhang et al., 2022; Xue et al., 2023). GPX4 is a key regulatory factor that reduces the lipid peroxidation level of tumor cells and inhibits the activities of cyclooxygenase and lipoxygenase (Wang et al., 2022b; Li et al., 2023; Ding et al., 2021; Zhang et al., 2023). Studies have shown that NAT10 promotes the progression of various malignant tumor diseases, such as colorectal, breast, and cervical cancer (Liu et al., 2019; Liu et al., 2020; Chen et al., 2023b; Jin et al., 2022). However, it remains unclear whether NAT10 promotes ccRCC progression by modulating ferroptosis. Therefore, we investigated this potential link. First, we assessed ferroptosis in clear-cell renal cell carcinoma tissues. The results indicated that the ferroptosis marker proteins in ccRCC tissues were significantly decreased, whereas the GPX4 and NFE2L1 proteins that inhibit ferroptosis were significantly increased, suggesting that ferroptosis in ccRCC was significantly suppressed. Recent studies have found that NAT10 significantly influences the intracellular iron content and expression of the ferroptosis metabolite MDA in tumor cells (Emdad et al., 2020; Tang et al., 2024; Emad et al., 2023 Chu et al., 2023). To explore the relationship between NAT10 and ferroptosis, we silenced and overexpressed NAT10 in ACHN and Caki1 cells, respectively. The results showed that when NAT10 was overexpressed, the levels of ferroptosis marker proteins in ACHN and Caki1 cells significantly decreased. Conversely, when NAT10 was inhibited, the levels of ferroptosis marker proteins notably increased. Combined with the above-mentioned effects of silencing and overexpression of NAT10 in ACHN and Caki1 cells on the malignant biological behavior of ccRCC, it was suggested that in ccRCC, NAT10 could inhibit ferroptosis and promote disease progression. Research on the relevant molecular mechanism by which NAT10 regulates ferroptosis revealed that NAT10 affected ferroptosis, which is closely related to tumor cells, by influencing the key regulatory molecules GPX4 and NFE2L1 of ferroptosis (Waku et al., 2020). We also carried out this study, and the results indicated that after silencing NAT10 in ACHN and Caki1 cells, the expression of GPX4 and NFE2L1 was significantly reduced, while after overexpression of NAT10, the expression of GPX4 and NFE2L1 was significantly elevated. This demonstrated that GPX4 and NFE2L1 were significantly regulated and influenced by NAT10. Current studies on ferroptosis have shown that modulating ferroptosis marker proteins can affect the progression of various malignant tumors, coronary artery atherosclerosis, and neurodegenerative diseases (Koppula, Zhuang & Gan, 2021; Chen et al., 2023a; Shen et al., 2022; Zhou et al., 2021; Zhu et al., 2024), and ferroptosis may be a crucial direction for the clinical treatment of malignant tumors. Therefore, in this study, we evaluated the effects of NAT10 on the ferroptosis marker proteins PTGS2, and CHAC1 in clear cell RCC by regulating the NFE2L1-GPX4 pathway. The results demonstrated that when ACHN and Caki1 cells were treated with the NAT10-specific inhibitor HBr, the expression of GPX4 was suppressed. After the overexpression of NFE2L1, the inhibitory effect of HBr on GPX4 was partially alleviated. The CCK-8 assay, the level of apoptotic iron metabolites MDA and iron content in cells, as well as ferroptosis marker proteins PTGS2, and CHAC1 also showed the same trend. This indicates that HBr can promote ferroptosis in ccRCC by inhibiting the expression of NAT10, whereas NFE2L1 partially mitigates this promoting effect.

Limitation

Although this study identified NAT10 as a novel factor that inhibits ferroptosis in renal clear cell carcinoma via the NFE2L1-GPX4 axis, there are some limitations that need to be acknowledged: (a) Clinical sample limitations: our cohort included 30 paired pairs of renal clear cell carcinoma samples with limited clinicopathological stratification (e.g., only 35% stage III–IV and 20% high-grade tumors). Information such as key features (e.g., VHL mutation status critical for iron metabolism in renal clear cell carcinoma) and treatment history are not adequately labeled, which may interfere with the analysis of markers of ferroptosis. The sample size limits the multivariate prognostic analysis of the clinical relevance of NAT10. (b) Technical considerations: the evaluation of ferroptosis relies primarily on indirect markers (malondialdehyde, iron, PTGS2) rather than direct lipid peroxidation imaging (e.g., C11-BODIPY assay). Lack of in vivo validation—xenograft models were able to elucidate the effect of NAT10 on ferroptosis in the tumor microenvironment. (c) Biological background: the acetylation-dependent regulation of NFE2L1 by NAT10 has not been explored. Potential interference with renal clear cell carcinoma (ccRCC) signature pathways remains to be explored.

Conclusion

In summary, our research revealed that NAT10, functioning as a tumor-promoting factor, suppresses ferroptosis in ccRCC via the NFE2L1-GPX4 signaling pathway, thereby facilitating the malignant progression of the disease. This study presents a novel potential research target for the pathogenesis of ccRCC and provides a new perspective for exploring the tumor-promoting mechanism of NAT10. In conclusion, the results of this study are undoubtedly significant for a deeper understanding and treatment of clear cell RCC.

Supplemental Information

Supplemental Information 1 MIQE checklist

Supplemental Information 2 Expression changes of NAT10 in renal clear cell carcinoma tissue

Supplemental Information 3 NAT10 promotes the malignant biology of renal clear cell carcinoma

Supplemental Information 4 Bioinformatics identification of ferroptosis-related genes and pathways enrichment associated with NAT10 in ccRCC

Supplemental Information 5 NAT10 regulates the expression of ferroptosis-related proteins

Supplemental Information 6 NAT10 regulates the expression of GPX4 and NFE2L1

Supplemental Information 7 NAT10 suppresses ferroptosis via the GPX4-NFE2L1 signaling pathway

Supplemental Information 8 GPX4 knockdown reverses the ferroptosis resistance induced by NAT10 overexpression

Supplemental Information 9 Graphical summary illustrating the proposed mechanism where NAT10 and NFE2L1 regulate ferroptosis-related genes (SLC7A11, PTGS2, CHAC1, GPX4) in ccRCC tumor cells, ultimately influencing tumor cell proliferation

We thank every colleague who contributed to this experiment.

Additional Information and Declarations

Competing Interests

Author Contributions

Human Ethics

Ethics

Data Availability

The authors declare there are no competing interests.

Chao Tan conceived and designed the experiments, performed the experiments, analyzed the data, prepared figures and/or tables, and approved the final draft.

Yang Wang performed the experiments, analyzed the data, authored or reviewed drafts of the article, and approved the final draft.

XinJie Zhang performed the experiments, prepared figures and/or tables, and approved the final draft.

ZiKang Li conceived and designed the experiments, authored or reviewed drafts of the article, and approved the final draft.

Shubo Chen conceived and designed the experiments, performed the experiments, analyzed the data, authored or reviewed drafts of the article, and approved the final draft.

The following information was supplied relating to ethical approvals (i.e., approving body and any reference numbers):

The collection of tumor specimens received written informed consent from patients and their families and was approved by the Xingtai People’s Hospital Ethics Committee (2023-035).

The following information was supplied relating to ethical approvals (i.e., approving body and any reference numbers):

The collection of tumor specimens received informed consent from patients and their families and was approved by the Xingtai People’s Hospital Ethics Committee (2023-035),

The following information was supplied regarding data availability:

The raw data is available in the Supplemental Files.

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
