# Peer review of "NAT10 inhibits ferroptosis and promotes the progression of renal clear cell carcinoma by regulating the NFE2L1-GPX4 signaling pathway"

_PeerJ, doi:10.7717/peerj.20224_

## Round 0.1 · original submission · Major Revisions

Please address concerns of the reviewers and amend manuscript accordingly.

**Language Note:** The review process has identified that the English language must be improved. PeerJ can provide language editing services - please contact us at [email protected] for pricing (be sure to provide your manuscript number and title). Alternatively, you should make your own arrangements to improve the language quality and provide details in your response letter. – PeerJ Staff

Reviewer 1 ·

Basic reporting

Although NAT10 has been extensively studied in renal cancer, the authors have not acknowledged this body of work. It is essential that the authors summarize relevant studies and clearly articulate the significance and novelty of their current investigation.

Experimental design

The rationale for investigating the role of NAT10 in regulating ferroptosis specifically in ccRCC is unclear. The authors should explain how they identified this research angle and provide supporting justification.

Validity of the findings

Given the complexity and diversity of ferroptosis regulatory pathways, it is not sufficiently explained why the study focuses exclusively on the NFE2L1-GPX4 axis. The authors should clarify their reasoning for selecting this specific pathway over others.

Additional comments

The study investigates the role of NAT10 in clear cell renal cell carcinoma (ccRCC), with a particular focus on its interaction with ferroptosis. Through some in vitro assays, the authors demonstrate that NAT10 overexpression promotes proliferation and migration of ccRCC cells while suppressing ferroptosis-related markers.

This manuscript presents some interesting data; however, the following issues warrant close attention:

1-Although NAT10 has been extensively studied in renal cancer, the authors have not acknowledged this body of work. It is essential that the authors summarize relevant studies and clearly articulate the significance and novelty of their current investigation.

2-The rationale for investigating the role of NAT10 in regulating ferroptosis specifically in ccRCC is unclear. The authors should explain how they identified this research angle and provide supporting justification.

3-Given the complexity and diversity of ferroptosis regulatory pathways, it is not sufficiently explained why the study focuses exclusively on the NFE2L1-GPX4 axis. The authors should clarify their reasoning for selecting this specific pathway over others.

·

Basic reporting

In the manuscript entitled “NAT10 inhibits ferroptosis and promotes the progression of renal clear cell carcinoma by regulating the NFE2L1–GPX4 signaling pathway,” the authors investigate the role of NAT10 in clear cell renal cell carcinoma (ccRCC), highlighting its function in promoting tumor progression by inhibiting ferroptosis through the NFE2L1–GPX4 axis. The study combines data from clinical samples, cell lines, and multiple functional assays, and presents a potentially interesting regulatory mechanism with therapeutic relevance.

After a careful read, I find the study generally well-organized and the findings of potential interest. However, several issues need to be addressed before the manuscript can be considered for publication:

1. The manuscript would benefit from careful language editing. There are grammatical errors and awkward phrasings in several sections (e.g., abstract, introduction, and discussion) that affect readability.

2. For reproducibility, original data for Western blot, qPCR, and key assays should be made available as supplementary files or in a public repository, if not already provided.

3. Ensure consistent formatting of abbreviations throughout, and consider including a list of abbreviations for clarity.

4. A few more recent references (e.g., 2023–2025) on ferroptosis regulation in ccRCC would enhance the contextual background and discussion.

Experimental design

5. The resolution of certain images is suboptimal. Please revise for clarity. Also, ensure that statistical methods (e.g., replicates, tests used) are clearly stated in the figure legends or methods section.

6. The use of 30 paired clinical samples is appreciated, but it would be helpful to discuss their clinicopathological characteristics in more detail. Also, consider addressing the potential limitations of the small sample size.

Validity of the findings

7. The study proposes that NAT10 acts via NFE2L1 and GPX4 to suppress ferroptosis, but the causal relationship remains insufficiently validated. It is strongly recommended to include rescue experiments (e.g., GPX4 knockdown in NAT10-overexpressing cells) to strengthen mechanistic support.

Additional comments

The manuscript addresses a relevant and timely topic and provides some promising data. However, additional mechanistic validation, improved clarity of presentation, and careful editing are needed. I recommend major revision.

Reviewer 3 ·

Basic reporting

The article by Tan et al. argues that NAT10 promotes the malignant behavior of ccRCC and inhibits ferroptosis markers. While the manuscript uses clinical material for testing their predictions and utilizes assays based on well-established ferroptosis markers, it falls short of establishing direct evidence that NAT10 promotes malignant behavior by inhibiting ferroptosis. Because a direct measurement of ferroptosis is missing, the claim that NAT10 supports malignant cRCC can be made for several other growth inhibitory mechanisms, such as evasion of apoptosis, other stresses such as ER stress, oxidative stress, etc.

The manuscript needs improvement in clarity, grammar, and scientific writing. At several instances, an incorrect technical term is used. The method section needs improvement as it does not document the methodology for several types of measurements. Images need to be of better quality and appropriately marked where needed.

Experimental design

The manuscript by Tan et al. uses a reasonable experimental design but needs to include experiments that validate the original claim of NAT10 inhibiting ferroptosis and provide an improved methodology for scoring several of the measurements. It needs to incorporate assays that directly measure ferroptosis in cell culture experiments.

1. Authors need to provide their scoring and normalization methods for IF and IHC experiments. For example, in its current form, it is unclear what specific cell type is being scored for NAT10 staining in renal tissue. How are the cell numbers determined for IF staining, and how are the scores normalized? How many total cells and cells/image were scored?
Images need to be marked properly for the use of pseudocolor, and the size marker needs to be displayed.

2. Technical language needs to be improved. For example, in this sentence, it is unclear how the cells can be transduced after fixation.
"106 The tissue was fixed with a pre-cooled mixture of methanol and acetone (1:1) and then
107 transduced with 1% TritonX-100/PBS."

3. What is the need for refixation in this protocol?
In the same protocol, what is the purpose of refixation?
The next day, the cells were fixed with 4% paraformaldehyde for 4 h, and co-
110 incubated with PBS containing 15% sucrose for 4 h. The glass slides were cleaned with
111 deionized water, and the slices were analyzed using laser confocal scanning."

4. What plasmids were used for genetic engineering? The method section in the abstract states that viral vectors were used and stable cell lines were created, but the method in the manuscript does not mention this; rather only describes the use of the transfection protocol.

5. Crucial details of several assays are missing. How did the authors measure iron, MDA, and ROS?

Validity of the findings

1. For shRNA constructs, please provide shRNA sequences that were used to knock down NAT10.
ShRNA knockdown results need to be controlled for untargeted gene knockdown.

2. Can the effect of NAT10 knockdown be rescued by overexpressing an RNAi-resistant cDNA of NAT10 in the same cell line?

Additional comments

1. Fig. 1 G mentions time on the x-axis but does not mention if it is months, days, or years. \

2. Fig. 2B-C should be complemented with protein expression level. It is not clear why two overexpression plasmids are used. Are these different clones? If yes, how were the clones derived?

3. Figure 2 has a typo- invation

4. Figure 4 mentions western blot measurements of GPX4 and NFE2L1, but does not mention which cell lines.

5. Invasion assays in Fig. 2 were performed where NAT10 was downregulated or overexpressed in different cell lines. Why not show them both in the same cell line?

6. Fig. 1A, it would be great to point out the histological structure of renal tubules, which will help the reader.

---

## Round 0.2 · Minor Revisions

Please address the remaining concerns of Reviewer #2 and provide the required information. Please amend the manuscript accordingly.

·

Basic reporting

The manuscript is clearly written and well structured. Language has been improved since the first round. Background information is sufficient, with updated references. Figures and tables are appropriate and well presented.

Experimental design

The research question is clear and relevant. Experimental methods are appropriately designed and sufficiently detailed for replication. Rescue experiments and quantitative analyses were added in the revision, enhancing rigor. Ethical approval and sample sources are clearly described.

Validity of the findings

The data are robust and well controlled. The addition of GPX4 inhibition experiments strengthens the causal interpretation. Conclusions are supported by the results and aligned with the stated hypothesis.

Additional comments

This is a timely and well-revised study exploring the regulatory role of NAT10 in ferroptosis in ccRCC. The authors have addressed prior concerns with added experiments, clearer methods, and improved clarity. I have no further suggestions.

Reviewer 3 ·

Basic reporting

The manuscript has improved significantly compared to the original submission, and most of my concerns are addressed.
Below are my comments on the revised submission-
1. Figure legend 1A. Need to include information in figure legends-
What do the arrows and box represent?

2. Details of plasmid vectors are still missing.

Experimental design

No comments

Validity of the findings

No comments

Additional comments

No additional comments.

---

## Round 0.3 · accepted · Accept

All remaining concerns of the reviewer were addressed, and the revised manuscript is acceptable now.